# Sardinian Infants of Diabetic Mothers: A Metabolomics Observational Study

**DOI:** 10.3390/ijms241813724

**Published:** 2023-09-06

**Authors:** Angelica Dessì, Alice Bosco, Flaminia Cesare Marincola, Roberta Pintus, Giulia Paci, Luigi Atzori, Vassilios Fanos, Cristina Piras

**Affiliations:** 1Department of Surgical Sciences, University of Cagliari and Neonatal Intensive Care Unit, AOU Cagliari, 09042 Cagliari, Italy; alicebosco88@gmail.com (A.B.); gomberta@icloud.com (R.P.); vafanos@tiscali.it (V.F.); 2Department of Chemical and Geological Sciences, University of Cagliari, Cittadella Universitaria, SS 554, km 4.5, Monserrato, 09042 Cagliari, Italy; flaminia@unica.it; 3Department of Biomedical Sciences, University of Cagliari, Cittadella Universitaria, SS 554, km 4.5, Monserrato, 09042 Cagliari, Italy; giuliapeace94@gmail.com (G.P.); l.atzori@unica.it (L.A.); cristina.piras@unica.it (C.P.)

**Keywords:** metabolomics, GDM, newborn

## Abstract

Gestational diabetes mellitus (GDM) is a condition characterized by glucose intolerance, with hyperglycemia of varying severity with onset during pregnancy. An uncontrolled GDM can lead to an increased risk of morbidity in the fetus and newborn, and an increased risk of obesity or developing type 2 diabetes, hypertension or neurocognitive developmental impairment in adulthood. In this study, we used nuclear magnetic resonance (NMR) spectroscopy and gas chromatography–mass spectrometry (GS-MS) to analyze the urinary metabolomic profile of newborns of diabetic mothers (NDMs) with the aim of identifying biomarkers useful for the monitoring of NDMs and for early diagnosis of predisposition to develop related chronic diseases. A total of 26 newborns were recruited: 21 children of diabetic mothers, comprising 13 in diet therapy (NDM-diet) and 8 in insulin therapy (NDM-insulin), and 5 control children of non-diabetic mothers (CTR). Urine samples were collected at five time points: at birth (T1), on the third day of life (T2), one week (T3), one month (T4) and six months postpartum (T5). At T1, variations were observed in the levels of seven potential biomarkers (acetate, lactate, glycylproline/proline, isocitrate, N,N-dimethylglycine, N-acetylglucosamine and N-carbamoyl-aspartate) in NMD-insulin infants compared to NDM-diet and CTR infants. In particular, the altered metabolites were found to be involved in several metabolic pathways such as citrate metabolism, glycine, serine and threonine metabolism, arginine and proline metabolism, amino sugar and nucleotide sugar metabolism, and pyruvate metabolism. In contrast, these changes were not visible at subsequent sampling times. The impact of early nutrition (maternal and formula milk) on the metabolomic profile was considered as a potential contributing factor to this finding.

## 1. Introduction

Hyperglycemia of pregnancy (HIP) is one of the most common metabolic alterations in pregnancy [1]. It has been defined by the World Health Organization (WHO) as “diabetes first detected at any time during pregnancy” and includes both diabetes in pregnancy (DIP) and gestational diabetes (GDM). The high prevalence and incidence of diabetes in women of childbearing age has become one of the main research topics in perinatal medicine due to the possible link between maternal hyperglycemia and the short- and long-term consequences for both the mother and offspring [2]. While DIP is diagnosed using the same criteria as the general population, relying on the correlation between plasma glucose levels and the potential microvascular diabetes complications, GDM diagnosis is more complex [1]. Indeed, the criteria for GDM diagnosis are related to the risk of adverse outcomes in pregnancy. Given the close correlation between the increase in blood glucose and severe negative consequences, the International Association of Diabetes and Pregnancy Study Groups (IADPSG) Consensus Panel has based the diagnostic values of GDM on the odds ratio of 1.75 for neonatal adverse outcomes (birth weight > 90th percentile, cord C-peptide > 90th percentile and neonatal percent body fat > 90th percentile) compared to the mean values for fasting plasma glucose and 1–2 h after oral glucose tolerance test [1].

According to the International Diabetes Federation Diabetes Atlas [3], the global standardized prevalence of GDM for 2021 was 14%, based on the IADPSG criteria. Data from the same organization showed that, in 2019, 15.8% of live births worldwide were affected by some form of glycemic alteration during pregnancy, with GDM responsible for 83.6% of cases, while 8.5% were caused by type 1 or type 2 diabetes diagnosed for the first time in pregnancy [4]. Furthermore, the Hyperglycemia and Adverse Pregnancy Outcome (HAPO) Study has shown an important linear association between poor glycemic regulation and short-term adverse effects in pregnancy [5], including: fetal macrosomia [6], perinatal death, congenital anomalies [7,8], neonatal asphyxia, jaundice and kernicterus, neonatal respiratory distress syndrome [8], brachial plexus injury and shoulder dystocia [9]. Polycythemia, hypocalcemia and hypertrophic cardiomyopathy are also documented [9,10]. Ongoing research in the field has also highlighted important long-term repercussions, including glucose and insulin resistance in childhood, regardless of the child’s BMI and family history of diabetes [11], and a higher rate of overweight and obesity in offspring [12,13], as well as higher blood pressure levels [13,14] and a higher incidence of cardiovascular problems from adolescence to adulthood [15]. Moreover, long-term neuro-psychiatric adverse effects could also occur in the offspring [15]; there are also possible correlations with autism spectrum disorders [16,17] and ophthalmic problems such as the increased risk of high refractive error [18].

Metabolomics is a highly promising technique for the early detection of various fetal, perinatal, pediatric and adulthood conditions [19,20]. It has the potential to aid in disease progression monitoring, optimize therapy and evaluate associated side effects, with the goal of personalized management. In prenatal and perinatal metabolomics analysis, samples can be obtained from various sources such as mothers (for instance, amniotic fluid, placenta, blood, urine and breast milk) or from the neonate (urine, blood, saliva and stools), offering a snapshot of the metabolic state of the individual useful for the identification of potential biomarkers with a predictive and diagnostic value [19,20].

The objective of this study was the analysis of the urinary metabolic profile of newborns of diabetic mothers (NDMs) at different time points, in order to characterize possible metabolic alterations in the infant, that could also depend on maternal therapy, whether diet or insulin. The clinical application of these preliminary results, in the future, could be an earlier diagnosis of susceptibility to developing related chronic diseases in these infants. In addition, comparison with data in the literature may expand the knowledge, nowadays still scarce, regarding possible new biomarkers for monitoring NDM. This study was conducted through the combined use of ^1^H NMR spectroscopy, mass spectrometry (MS) and multivariate statistical analysis techniques.

## 2. Results

Multivariate data analysis was employed to examine the NMR spectral profile of urine samples. The identification of metabolites was guided by existing literature [21] as well as dedicated resources like the Human Metabolome Database (HMDB, http://www.hmdb.ca, accessed on 23 May 2023) and the 500 MHz library within the Chenomx NMR suite 7.1. The major NMR resonances originated from various molecular classes including free amino acids, organic acids, sugars, small organic compounds and osmolytes. A Principal Component Analysis (PCA) of the whole NMR dataset was performed to evaluate possible clustering (NDM-insulin, NDM-diet and CTR) and detect outliers lying beyond the 95% confidence limit. No samples necessitated exclusion as outliers. To highlight metabolic differences between NDM-insulin, NDM-diet and CTR, a supervised OPLS-DA analysis was subsequently executed for each sampling time point. Figure 1a shows the score plot of the model built with samples collected at T1. As can be noted, the NDM-insulin group was clearly separated from the other two groups, indicating the presence of distinct metabolic profiles. In contrast, the close proximity between NDM-diet and CTR subjects suggested a similar urine metabolome and therefore they were grouped together, defining a single class. An OPLS-DA model was built using two components (1P + 1O), showing favorable R2X, R2Y and Q2 values as outlined in Table 1. To validate the model, a permutation test was conducted (Figure 1b) through 400 iterations. The outcomes of this test, reported in Table 1, affirm the statistical reliability of the model. The spectral regions differing in NDM-insulin subjects compared to NDM-diet and CTR were identified in the S-line plot. A|p(corr)| > 0.6 was selected as the significance level. The results showed that the levels of acetate, glycylproline, isocitrate, N,N-dimethylglycine, N-acetylglucosamine and N-carbamoylaspartate in NDM-insulin subjects changed significantly compared to CTR and NDM-diet. Additional insights into these variations were obtained by comparing the relative contents of these metabolites using the Mann–Whitney U test with a Benjamini–Hochberg correction. As shown in Figure 2, NDM-insulin subjects were characterized by significantly higher levels of glycylproline, isocitrate and N,N-dimethylglycine, and lower levels of acetate, N-acetylglucosamine and N-carbamoylaspartate compared to CTR + NDM-diet. Similar findings were obtained by the analysis of GC-MS data. Approximately 46 metabolites were identified. The outcomes of the univariate statistical analysis of metabolite levels revealed that only at time T1 NDM-insulin subjects were characterized by different metabolite levels compared to CTR + NDM-diet. Specifically, the NDM group exhibited higher contents of proline and N,N-dimethylglycine, and lower levels of acetate, N-acetylglucosamine and lactate compared to CTR + NDM-diet (Figure 3). Metabolomics plays a crucial role in screening potential biomarkers for clinical diagnoses. The evaluation of diagnostic accuracy for these biomarkers has been successfully conducted in numerous studies using receiver operating characteristic (ROC) curves. A biomarker’s diagnostic accuracy can be determined by its area under the ROC curve (AUC) value, with an AUC between 0.9 and 0.7 indicating a certain level of accuracy and an AUC above 0.9 indicating exceptionally high accuracy. In our study, we employed ROC analysis to evaluate the sensitivity and specificity of the combination of metabolites suggested by the NMR and GC-MS analysis of urine samples at T1. Figure 4 shows the corresponding ROC curves which exhibited an AUC of 0.894 (95% CI: 0.672–1) and 0.925 (95% CI: 0.700–1) for NMR and GC-MS, respectively. These findings suggested a good predictive accuracy of the potential combinatorial biomarker. Finally, a pathway analysis incorporating all the significantly different metabolites (NMR + GC-MS) was conducted. As shown in Figure 5, citrate metabolism, glycine, serine and threonine metabolism, arginine and proline metabolism, amino sugar and nucleotide sugar metabolism, and pyruvate metabolism were the most altered pathways between the NDM-insulin subjects and the joint CTR + NDM-diet group. To evaluate the changes in insulin treatment on the urinary metabolome of newborns at T1, an OPLS-DA model excluding controls was performed on the ^1^H NMR data. As expected, Figure 6 shows a good separation along t [1] between NDM-insulin and NDM-diet subjects. The OPLS-DA model was built using a component (1P) and the statistical values are reported in Table 1. The result of the permutation test is reported in Table 1 and indicates the validity of the OPLS-DA model. Differently from what was observed when the controls were included in the model, only the glycylproline (*p*-value = 0.02), isocitrate (*p*-value = 0.05) and N-acetylglucosamine (*p*-value = 0.0009) discriminate between the two groups of patients, showing higher levels in glycylproline and isocitrate, and low levels of N-acetylglucosamine in NDM-insulin subjects. Univariate analysis performed on the GC-MS data shows that lactate, proline and N-acetylglucosamine retains both trend and statistical significance (*p*-value = 0.03).

## 3. Discussion

In this study, the urinary metabolome of NDM-insulin infants was found to be significantly different from that of NDM-diet and control infants, in samples taken within 12 h after birth (T1). On the other hand, no significant differences were observed in samples obtained at 4, 7 and 30 days after birth, possibly due to differences in infant feeding and their effects on the urine metabolome of newborns [22]. Indeed, NDM-insulin infant urine samples collected at T1 were characterized by higher levels of glycylproline/proline, isocitrate and *N,N*-dimethylglycine (DMG), and lower levels of acetate, *N*-acetylglucosamine, *N*-carbamoylaspartate and lactate compared to NDM-diet and controls. These results are similar to those reported by previous studies on gestational diabetes [23,24].

The increase in DMG observed in the NDM-insulin group could represent an initial metabolic alteration reflecting the impact of poor glycemic compensation on cardiovascular risk. In the literature, several studies support this hypothesis, including one conducted by Andraos et al. [25]. They analyzed a cohort of Australian children and adults, in which they highlighted the association between the concentration of some trimethylamine N-oxide (TMAO) precursors, including DMG, with metabolic syndrome, cardiometabolic and adverse inflammatory phenotypes. Moreover, in a previous study [26] conducted in an adult population, high values of plasmatic DMG were correlated with the risk of mortality in patients with suspected or confirmed coronary artery disease. This relationship was greater for deaths from cardiovascular causes than for non-cardiovascular ones [26]. Furthermore, plasma DMG was independently related to acute myocardial infarction and improved risk prediction in patients with stable angina pectoris [27]. Indeed, DMG is a tertiary amine that originates from betaine in the process of re-methylation of homocysteine into methionine, by betaine-homocysteine methyltransferase (BHMT), mainly localized in the liver and kidney. BHMT is responsible for blood levels’ regulation of DMG, with repercussions for hepatic concentrations of S-adenosylmethionine and for the availability of methyl groups for transmethylation reactions. Among these, it is also involved in the synthesis of phosphatidylcholine, the main phospholipid in very low-density lipoprotein particles, with possible repercussions also for the assembly of circulating plasma lipoproteins [27].

The increase in proline that characterized the urine samples of children born to diabetic mothers on insulin therapy compared to the diet therapy group and controls was statistically significant, as well. Interestingly, the increase in this metabolite was significant even when NDM-insulin infants’ urine metabolome was compared to that of NMD-diet infants only. This increase could reflect altered fetal metabolism or indicate improper placental exchange of amino acids although the correlation with maternal proline values is not yet clear [23,28,29,30,31,32,33]. In fact, Scholtens et al. [28] found that, out of 117 women, those with high fasting glycemic values had a serum metabolic profile consistent with insulin resistance, which included high proline values, compared to controls. Graca et al. [23] showed a decrease in proline levels in the amniotic fluid in the pre-diagnostic period (14–25 weeks of gestation) in women with gestational diabetes diagnosed after 24–28 weeks of gestation. Furthermore, the increase in proline has been discussed in a recent study of the umbilical cord blood of children born to mothers with gestational diabetes compared to control children [29]. In this study, Lu et al. [29] found higher concentrations of ten amino acids in the cord blood of infants of mothers with GDM, compared to infants of non-GDM mothers, in the absence of alteration of the maternal metabolome between mothers with GDM and controls. However, only fetal proline showed an independent association with GDM. Proline was also related to gestational age [29]. Indeed, the multivariable regression models highlighted the possibility that the high proline levels in the umbilical cord blood of the GDM offspring could affect the gestational age, which could be low in case of maternal diabetes. Therefore, they highlighted an active role of the fetus, which would not be only passively affected by this condition. Furthermore, previous publications confirmed a possible negative correlation between proline in amniotic fluid, neonatal blood and gestational age [30,31]. The finding of high proline values is therefore a confirmation of the results of Cetin et al. [32], the first group that analyzed fetal and maternal amino acid concentrations in the presence of GDM. In fact, they detected high concentrations of proline in the cord blood, while this amino acid was absent in the maternal circulation. Indeed, some recent studies seem to confirm the decrease in proline values as a characteristic of mothers affected by GDM, even before the diagnosis [33,34].

Higher levels of isocitrate in NDM-insulin children are related to an alteration, potentially induced by fetal malnutrition, of the tricarboxylic acid cycle (TCA). High concentration of this metabolite was also discriminant between NDM-insulin children and NMD-diet children, when we compared these two groups excluding the control group. This condition is a pathophysiological process associated with the diabetic phenotype, as also observed in other studies [35], even before clinical evidence of pathology [36]. Indeed, Gaster et al. [35] showed how the reduction of the TCA cycle turns out to be both a marker of and responsible for the diabetic phenotype. It has already been found that in the muscle tissue (myotubes) in case of diabetes there is a reduced primary flow of the TCA cycle, a condition that also characterizes the insulin-resistant offspring of patients with type 2 diabetes. Befroy et al. [36] started from the analysis of young, lean and healthy individuals to eliminate confounding factors, monitoring their glucose tolerance over time in order to isolate the first metabolic alterations associated with insulin resistance. They found that in subjects who later developed impaired glucose tolerance there was a TCA cycle flow deficit in muscle prior to the onset of the impairment. In addition, Scholtens et al. [28] showed elevated isocitrate levels in pregnant women with elevated fasting glucose levels.

Furthermore, the alteration of the intermediates of TCA or metabolites related to it in the offspring of diabetic mothers was detected in other metabolomics studies on urine and cord blood [37,38,39]. A potential correlation between the alteration of the metabolites of the TCA cycle and fetal malnutrition was also highlighted by Cesare Marincola et al. [40]. In fact, they found that the alterations of the metabolites involved in the Krebs cycle were greater in infants with intrauterine growth retardation (IUGR) and large for gestational age (LGA) compared to AGA (adequate for gestational age), with incremental values from birth to the seventh day of life. The correlation of these data with those of Dessì et al. [41] suggests that these alterations may derive from a potential alteration in the fetal nutrition to which they were subjected, given the poor influence of nutrition on the neonatal metabolome at birth (T1).

Further confirmation of a possible negative association between the alteration of the metabolites involved in the TCA cycle and health status was highlighted by Cheng et al. [42]. In this study, high isocitrate concentrations were negatively correlated with longevity (defined as reaching 80 years of age) and were associated with worse cardiovascular health, as well as risk of cardiovascular disease and death. Moreover, isocitrate is involved in the NAD(P)H production H, essential for the [43] cellular antioxidant defense system. Therefore, its increase could also be interpreted as the initial response to an increase in oxidative stress related to a non-optimal glycemic compensation. In addition, Guasch-Ferré et al. [44] showed how isocitrate is one of the metabolites significantly associated with the risk of type 2 diabetes. Furthermore, Ferdaoussi et al. [44] provided a possible further correlation of isocitrate with glucose metabolism, showing that the generation of NADPH dependent on cytosolic isocitrate dehydrogenases and the subsequent reduction of glutathione (GSH) contribute to the amplification of insulin exocytosis through sentrin/SUMO-specific protease-1 (SENP1).

Thus, these concentrations in the offspring of insulin-treated diabetic mothers could be interpreted as an attempt to potentiate the insulin response, although a decrease in insulin response has been identified as a potential index of renal impairment in the early stage of diabetic renal disease [45]. However, the role of isocitrate in the regulation of insulin secretion is still to be defined more clearly. In fact, an animal model study by Baucle et al. [46] showed that the mitochondrial citrate-isocitrate transporter (CIC) is not required for glucose-stimulated insulin secretion, and that additional complexities exist due to the role of cytosolic isocitrate dehydrogenase (Idh1) and NADPH in regulating β-cell function. Indeed, Idh1 inhibited insulin secretion in wild-type islets but had no impact on β-cell function in knockout CIC islets.

The decrease in the levels of *N*-acetylglucosamine observed in NDM-insulin infants could reflect a major dysregulation of glucose availability that characterizes NDM-insulin infants in an early stage, preceding insulin resistance. Moreover, the decrease in this metabolite was characteristic of the NDM-insulin infants even when only compared to NDM-diet infants. Indeed, this metabolite is related to the O-GlcNAc glycosylation process (O-GlcNAcylation), strongly correlated to the cellular nutritional state depending on the availability of glucose [47]. O-GlcNAcylation is a dynamic post-transcriptional modification, similar to phosphorylation, responsible for regulating the stability, activity or subcellular localization of target proteins, that consists in the addition of N-acetylglucosamine on the serine and threonine residues of cytosolic and nuclear proteins [47]. In their literature review, Issad et al. [47] stated that the level of O-GlcNAc on proteins is regulated only by the activity of two enzymes: O-GlcNAc transferase (OGT) and O-GlcNAc-ase (OGA), respectively responsible for the addition and removal of N-acetylglucosamine. This condition determines the impossibility for the O-GlcNAcylations to organize themselves into cellular signaling cascades typical of phosphorylations, acting as regulators of the intensity of signals in numerous metabolic pathways as a function of the nutritional state of the cell. For example, OGT attenuates insulin signaling through O-GlcNAcylation of proteins involved in the proximal and distal phases of the phosphatidylinositol-*3* (PI-3) kinase signaling pathway. This mechanism, in case of high cellular concentrations of glucose, seems to be related to the alterations of the insulin signal observed in diabetic patients, but also to the excessive production of glucose by the liver and to the deterioration of the pancreatic β-cell function, with a consequent worsening of hyperglycemia and the subsequent glucotoxicity. Furthermore, O-GlcNAcylations appear to be associated with various diabetic complications. Indeed, at the endothelial level they appear to be involved in micro- and macrovascular complications, while as regards vascular and renal dysfunctions they appear to activate the expression of profibrotic and antifibrinolytic factors. Recent studies [48,49] have shown that serum GlcNAc levels in humans seem to increase with insulin resistance. It has also been reported that the intake of GlcNAc orally causes, in an animal model, a weight gain without increasing the caloric intake, suggesting a greater efficiency in the extraction of nutrients. This is supported by Abdel Rahman et al. [50] which demonstrated how, in cell cultures, GlcNAc and the concomitant overexpression of Mgat5 glycosyltransferases (alpha-1,6-mannosylglycoprotein 6-beta-N-acetylglucosaminyltransferase) determine a greater absorption of glutamine and essential amino acids. An increased responsiveness to [51] growth factors was also observed.

Moreover, the decrease in acetate in NMD insulin newborns, probably in the presence of a greater impairment of maternal glycemic control, may reflect the mounting evidence of an association between GDM and alterations in the intestinal bacterial flora in both mother and offspring. The most recent studies [52] in this regard seem to support the existence of specific alterations in the microbiota both in the mother and in the offspring, and the presence of similar trends of some altered bacterial species in the mother–child pair. This seems to suggest not only a possible form of intergenerational transmission but also an effect of the maternal microbiota on the gut bacterial colonization of the offspring [52]. However, the results are still controversial. On the one hand, Wang et al. [53] found that short-chain fatty acids (SCFA), produced by the intestinal microbiota from dietary fibers, including acetate, propionate and butyrate, were decreased in the GDM during the second and third trimesters, especially in subjects with greater impairment of glucose tolerance. This confirms the pre-diagnostic decrease in amniotic fluid acetate levels during the second trimester observed by Graca et al. [23] in women with GDM diagnosed after 24–28 weeks of gestation. Furthermore, a reduction in placental G protein-coupled receptors 41 and 43 (GPR41/43) and an increase in histone deacetylase (HDAC) were observed in GDM, together with an increased inflammatory response and impaired glucose metabolism at the placental level. Indeed, maternal circulating SCFAs, in particular acetic, propionic and butyric acids, seemed to be closely related to clinical markers in GDM pregnancies and may exert favorable effects on physiological activities either directly or through GPR and HDAC, affecting the placental immunometabolism and fetal development at birth [53]. Instead, Liu et al. [54] showed an increase in acetate levels both pre- and postpartum in mothers with GDM, although the greatest increases characterized mothers with GDM not treated with insulin.

Finally, the decrease in lactate levels, found only by the analysis with GC-MS, in NMD-insulin infants, differs from the data reported in the literature, making further evaluation necessary. In fact, Taricco et al. [55] analyzing fetuses of diabetic mothers showed increased umbilical glucose concentrations despite normal maternal glucose levels, and decreased oxygen saturation and O_2_ content together with increased lactate concentration, reflecting impaired fetal metabolism. Therefore, the authors hypothesized that “good maternal metabolic control” is not sufficient to ensure a normal metabolic environment for the fetus in GDM pregnancies. Furthermore, increased lactate levels also characterize women with GDM and this increase has also shown a significant positive correlation with HbA1c, blood pressure and fetal birth weight [56]. Moreover, Liu et al. [57] showed an increase in serum lactic acid levels in women with GDM compared to healthy controls.

The AUC values of our ROC curves (both ^1^H NMR and GC-MS) suggested a good predictive accuracy of the combined potential biomarkers. Nevertheless, from our data and according to the literature, the best candidates to be used as biomarkers, in the future, are proline and isocitrate. Indeed, it has been described how fetal proline has already shown an independent association with GDM, and alterations in TCA intermediates have already been revealed as characteristic of the urine and cord blood of the child of a diabetic mother, supporting their potential clinical relevance.

The presence of such alterations only in NMD-insulin infants compared to NDM-diet and CTR suggests that the greatest effects on the newborn occur in cases where the diabetic pathology is more complex and does not benefit from dietary correction alone, probably indicative of greater glycemic imbalances. In this regard, through studies with a higher number of patients, adequate specificity and sensitivity of these biomarkers can be identified, and then they could be used to highlight children potentially at risk of developing chronic non-communicable diseases in order to undertake as soon as possible an appropriate lifestyle to counteract this predisposition.

The main strengths of our work are the use of the dual methods of ^1^H-NMR and GC-MS analysis, and the AUC values of our ROC curves to evaluate the sensitivity and specificity of the metabolite combination suggested by the analysis.

There is, however, the limitation of the number of samples, which is quite small, and we therefore hope for future confirmation from studies with higher numbers of samples and patients.

## 4. Materials and Methods

### 4.1. Subject and Sample Collection

This study adhered to the Helsinki Declaration of 1975, with revisions made in 1983. The ethical committee approved the study protocol and written informed consent was obtained from the parents before enrolment in the study (CA-PG/2017/17959-29/12/2017). Urine collection was conducted in adherence with the ethical norms set forth by the responsible human experimentation committee.

Urine samples were obtained from 26 infants whose average gestational age was 38 ± 5 weeks. The sampling occurred within 12 h after birth (T1), at 4 (T2), 7 (T3) and 30 (T4) days postpartum, and 6 months (T5) after birth. Of these 26 children, 8 were born from diabetic mothers on insulin therapy (NDM-insulin), 13 to diabetic mothers on diet therapy (NDM-diet) and 5 were healthy controls.

A summary of the subjects and samples is reported in Table 2. The criteria for exclusion encompassed major malformations, congenital heart diseases, perinatal asphyxia, sepsis and need for surgery.

Urine samples from infants (2 mL each) were acquired through a non-invasive procedure involving the insertion of a cotton ball into a disposable diaper. The collected urine was then drawn into a syringe and transferred to sterile 1.5 mL vials. All specimens were subsequently preserved at −80 °C until the time of NMR analysis.

### 4.2. Urine Sample Preparation and ^1^H-NMR Analysis

An aliquot of 800 µL of urine was transferred into an Eppendorf tube with 8 µL of a 1% aqueous solution of NaN_3_ to inhibit bacteria growth and stored at −80 °C. Before the analysis, the sample was centrifuged at 12,000× *g* for 10 min at 4 °C to remove solid particles. Then, 630 µL of the supernatant was mixed with 70 µL of potassium phosphate buffer in D_2_O (1.5 M, pH 7.4) containing sodium 3-trimethylsilyl-propionate-2,2,3,3,-d_4_ (TSP) as an internal standard (98 atom% D, Sigma-Aldrich, Milan, Italy). Finally, an aliquot of 650 µL was transferred to 5-mm NMR glass tubes for ^1^H-NMR analysis. NMR analysis was carried out using a Varian UNITY INOVA 500 spectrometer operating at 499.839 MHz for proton and a 5 mm double resonance probe (Agilent Technologies, Santa Clara, CA, USA). The spectra were obtained employing a standard 1D Nuclear Overhauser Enhancement Spectroscopy (1D-NOESY) pulse sequence, which included pre-saturation of the water resonance, utilizing a mixing time of 1 ms and a relaxation delay of 3 s. For each spectrum, 256 free induction decays (FIDs) were collected into 64 K data points over a spectral width of 6000 Hz. The acquisition time was set to 2 s and the pulse to 90°. The FIDs were weighted by an exponential function with a 0.3 Hz line-broadening factor prior to Fourier transformation.

### 4.3. H NMR Data Preprocessing

NMR spectra were phased, and baseline corrected using an ACDlab Processor Academic Edition (Advanced Chemistry Development, 12.01, 2010) and chemical shifts referenced internally to TSP at d = 0.0 ppm. Next, the spectral range containing the residual water signal (4.7–4.9 ppm) was excluded. The final analyzed spectral regions were defined between 0.7–4.7 ppm and 4.9–9.5 ppm. Spectral integration was carried out using the ACD Labs intelligent bucketing method [58]. A bucket width of 0.04 ppm with a 50% looseness factor was used. This flexibility allows the bucket width to deviate slightly from the set value. The intelligent bucketing technique identifies local minima within the spectra and adjusts the bucket positions accordingly. This approach ensures that peaks are integrated into their respective buckets, even when minor chemical shift differences are present due to factors like pH variations. The areas of the bucketed regions were normalized using Median Fold Change Normalization (MFC) [59], a method particularly favored when working with urine samples compared to total sum normalization.

### 4.4. GC-MS Sample Preparation and Analysis

Blanks were made following the same procedure used for the samples to avoid noises due to the chemicals used for the preparation and the laboratory instruments. Derivatization was made by adding to dried samples 50 μL of methoxamine hydrochloride in pyridine solution (10 mg/mL) for 17 h. Subsequently, 50 μL of N-trimethylsilyltrifuoroacetamide (MSTFA) was added. Samples were then diluted in 600 μL of hexane with an internal standard (undecane at 25 ppm). Diluted samples were then transferred into glass vials. A 1 μL aliquot of the samples was injected splitless by an autosampler in an Agilent 7890 A gas chromatograph coupled with an Agilent 5975 C mass spectrometer equipped with a HP-5MS capillary column (5%-Phenyl-methylpolysiloxane; 30 m, 25 mm i.d., 0.25 μm film thickness). The initial oven temperature was 50 °C and increased at 10 °C/min to 250 °C for a total run of 35 min. The chromatograms were acquired in electron impact mode and full scan monitoring mode (*m*/*z* 50–800). The injector and ion source temperature were respectively set at 200 and 250 °C. Helium was used as the carrier gas in constant pressure mode (7.6522 psi). The metabolites were identified using the standard NIST 08 and Golm Metabolome Database (GMD) mass spectra libraries, as well as by comparison with authentic standards. The R library XCMS was used for peak detection and retention time correction. Parameters utilized for peak deconvolution for GC–MS matrices were manually optimized [60].

### 4.5. Multivariate Statistical Analysis

Multivariate statistical analysis (MVA) was performed using the SIMCA software (Version 16.0, Sartorius Stedim Biotech, Umea, Sweden), prior pre-processing of data by Pareto scaling [61]. A PCA was performed to detect potential trends or outliers [62]. OPLS-DA was conducted for a deeper analysis of discrimination among groups. The OPLS-DA models were constructed based on the internal 7-fold cross-validation. The variance and predictive ability of models were assessed by considering the R2X and Q2 parameters. R2X describes how much of the variation in the X variable could be explained by the selected components. Q2 describes how well the X could predict the Y. A prediction model is deemed effective when Q2 > 0.5. To validate the models, a permutation test with 400 iterations was conducted. This test involved shuffling the Y-matrix (representing class assignments or continuous variables), while keeping the X-matrix (denoting peak intensities in NMR spectra) unchanged. The permutation plot then illustrates the correlation coefficient between the original y-variable and the permuted y-variable along the *x*-axis, while the cumulative R2 and Q2 are plotted along the *y*-axis, accompanied by a regression line. The intercept of this line serves as a gauge of overfitting, with a Q2Y intercept value less than 0.05 indicating a valid model.

To identify the potential metabolites primarily responsible for group differentiation, an S-line plot was generated for the OPLS-DA model. The S-line plot is a tailored plot for NMR spectroscopy data, unifying the covariance (peak height) and correlation (color coding) of model variables within a single graph. Specifically, within the spectra, red signals corresponded to metabolites that made a more substantial contribution to the separation between the groups compared to blue signals. Additionally, the observed phase of the resonance signals on the predictive component indicated the reduction or elevation (negative or positive peaks) of metabolite levels in the respective groups.

### 4.6. Univariate Statistical Analysis

GraphPad Prism software (version 7.01, GraphPad Software, Inc., Boston, MA, USA) was employed for the univariate statistical analysis of the datasets. The significance of differences in metabolite concentrations was assessed using the Mann–Whitney U test, with a *p*-value < 0.05 indicating statistical significance. To account for multiple testing, the obtained *p*-values were subjected to the Benjamini–Hochberg adjustment [63] to determine the level of significance. Pathway analysis and receiver operating characteristic (ROC) analysis were performed using the MetaboAnalyst 5.0 (https://www.metaboanalyst.ca, accessed on 24 May 2023) program. The ROC analysis was conducted to evaluate the diagnostic robustness of potential biomarkers. The construction of the ROC curve was achieved using the Linear SVM algorithm.

## 5. Conclusions

This exploratory study contributes to the understanding of the biochemical mechanisms of GDM. Seven urinary metabolites (acetate, lactate, glycylproline/proline, isocitrate, *N,N*-dimethylglycine, *N*-acetylglucosamine and *N*-carbamoyl-aspartate) were found to contribute most to the separation of NMD-insulin infants from NMD-diet infants and the control group at 12 h postpartum. The absence of significant differences between the urine samples of the three groups up to 6 months after birth emphasizes the significant influence of nutrition (breast milk and infant formula) on the urinary metabolomic profile. Moreover, it underscores the potential of metabolomics to very promptly identify biomarkers useful for monitoring NMDs and for early diagnosis of predisposition to developing related chronic diseases. Indeed, specifically, the altered metabolites were related to several crucial metabolic pathways, including the metabolism of citrate, pyruvate, glycine, serine, threonine, arginine and proline as well as the metabolism of amino sugars and nucleotide sugars. Furthermore, the correlation of some metabolic alterations detected, such as those of proline, with recent data published in the literature brings out an active role of the fetus, no longer only passively affected by the gestational pathology. The results of this study can be considered a starting point for an individualized medicine in the future, with the elaboration of a nutrition specifically adapted to the individual needs of the children of diabetic mothers, with a view to reducing the long-term consequences of this pathology.

## Figures and Tables

**Figure 1 ijms-24-13724-f001:**
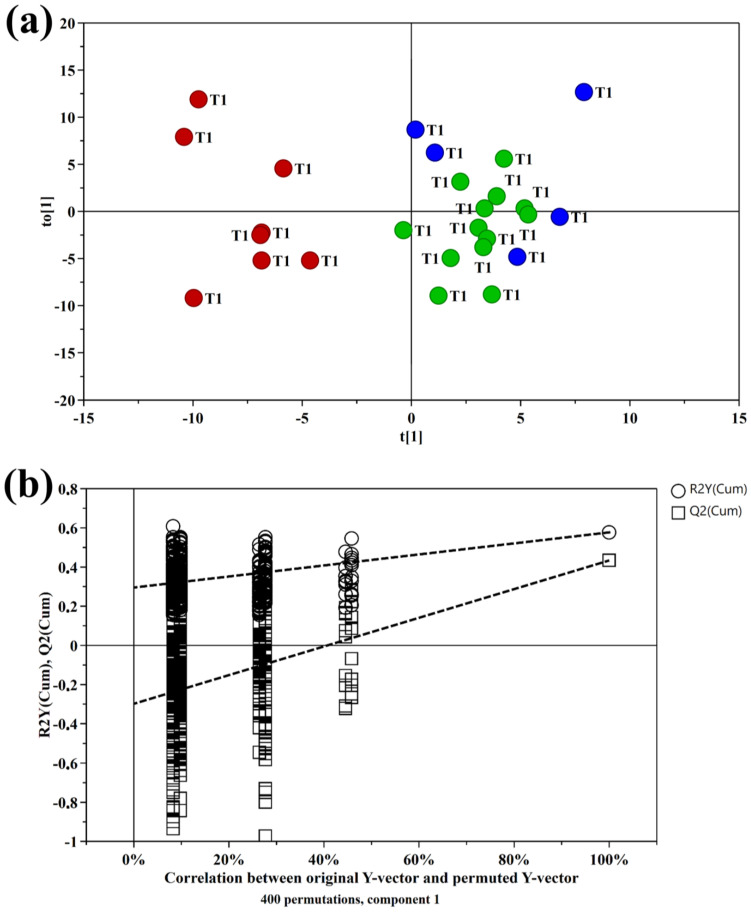
(**a**) Score plot of the OPLS-DA model built with the ^1^H NMR spectra of urine samples collected at time T1: controls (blue circle); NDM-diet (green circle); NDM-insulin (red circle). (**b**) Permutation plot from 400 permutation tests with Q2 intercept of −0.298. The horizontal axis shows the correlation between the permuted and actual data, while the vertical axis displays the cumulative values of R2 and Q2. The intercept gives an estimate of the overfitting phenomenon.

**Figure 2 ijms-24-13724-f002:**
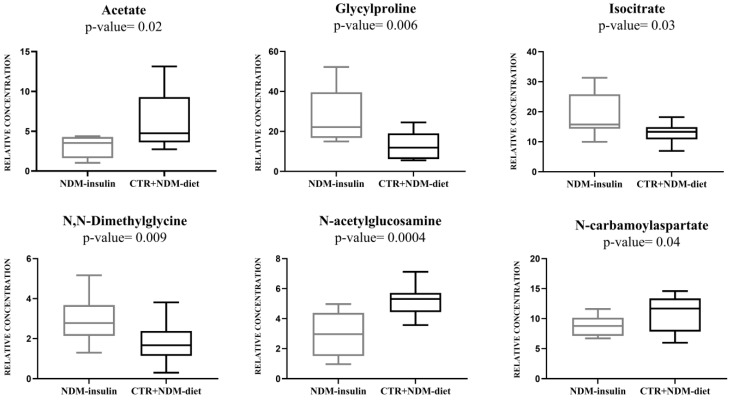
Box-and-whisker plots of the relative levels of metabolites in CTR + NDM-diet and NDM-insulin groups measured by ^1^H NMR technique. The Mann–Whitney U test was utilized to establish statistical significance, with a significance level set at *p*-value < 0.05. Furthermore, the Benjamini–Hochberg adjustment was implemented.

**Figure 3 ijms-24-13724-f003:**
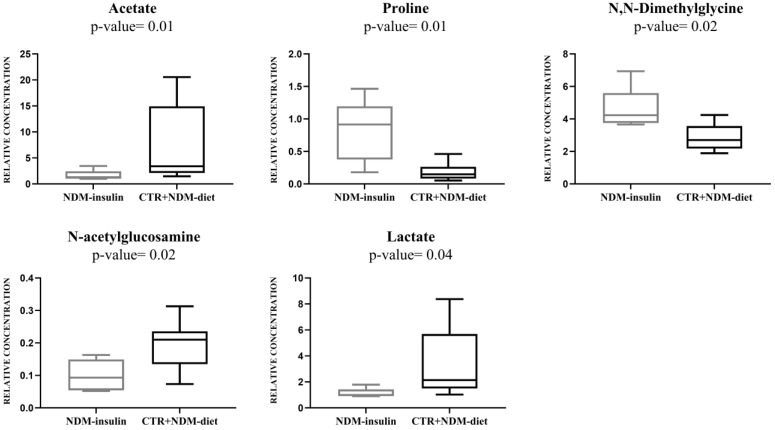
Box-and-whisker plots of the relative levels of metabolites in CTR + NDM-diet and NDM- insulin groups measured by GC-MS technique. The Mann–Whitney U test was utilized to establish statistical significance, with a significance level set at *p*-value < 0.05. Furthermore, the Benjamini–Hochberg adjustment was implemented.

**Figure 4 ijms-24-13724-f004:**
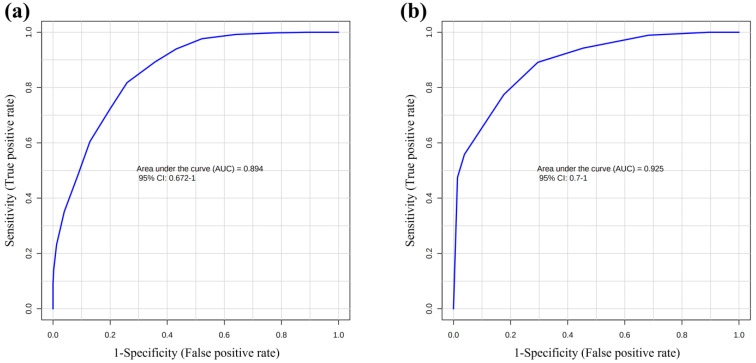
ROC curves built with metabolites with statistically significantly different levels between CTR + NDM-diet and NDM-insulin samples: (**a**) NMR; (**b**) GC-MS techniques. AUC = area under the curve.

**Figure 5 ijms-24-13724-f005:**
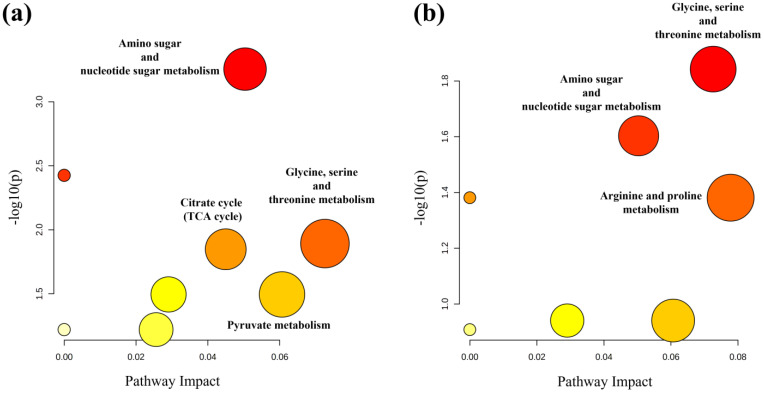
Metabolic pathway analysis of the set of metabolites found statistically different between CTR + NDM-diet and NDM by ^1^H NMR (**a**) and GC-MS (**b**) analysis of urine samples taken at time T1. The plots were generated using MetaboAnalyst 5.0. On the *X*-axis, the impact of the identified metabolites on the specified pathway is represented, while the *Y*-axis indicates the level of enrichment of the designated pathway by the identified metabolites. The color of the circles signifies the significance of pathway enrichment, whereas the circle size indicates the impact on the pathway.

**Figure 6 ijms-24-13724-f006:**
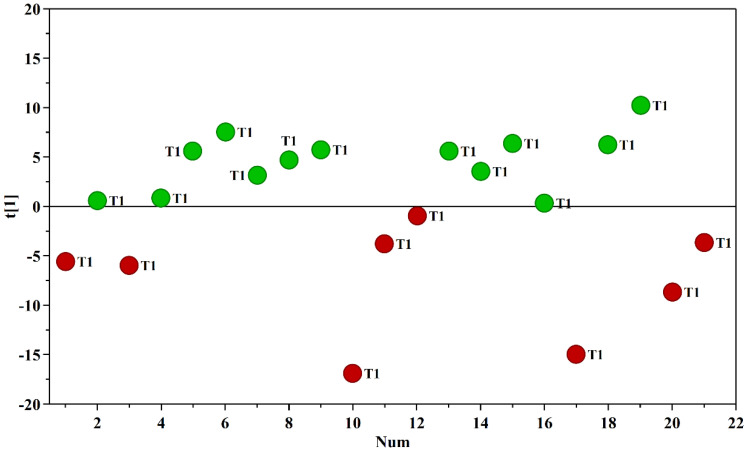
Score plot of the OPLS-DA model built with the ^1^H NMR spectra of urine samples collected at time T1: NDM-diet (green circle); NDM-insulin (red circle).

**Table 1 ijms-24-13724-t001:** Statistical parameters of OPLS-DA models.

	OPLS-DA Models				Permutation (400 Times) *	
	Components ^a^	R2Xcum ^b^	R2Ycum ^c^	Q2cum ^d^	R2 Intercept	Q2 Intercept
T1 NDM-insulin vs. NDM-diet and CTR	1P + 1O	0.383	0.858	0.507	0.295	−0.298
T2 NDM-insulin vs. NDM-diet and CTR	1P + 2O	0.721	0.680	0.183	-	-
T3 NDM-insulin vs. NDM-diet and CTR	1P + 1O	0.296	0.662	−0.546	-	-
T4 NDM-insulin vs. NDM-diet and CTR	1P + 1O	0.629	0.439	−1.010	-	-
T5 NDM-insulin vs. NDM-diet and CTR	1P + 2O	0.721	0.784	0.228	-	-
T1 NDM-insulin vs. NDM-diet	1P	0.296	0.694	0.572	0.304	−0.324

^a^ The number of predictive and orthogonal components used to create the statistical models. ^b,c^ R2X and R2Y indicate the cumulative explained fraction of the variation of the X block and Y block for the extracted components. ^d^ Q_2_ cum values indicate cumulative predicted fraction of the variation of the Y block for the extracted components. * An R2 intercept value less than 0.3–0.4 and a Q2 intercept value less than 0.05 are indicative of a valid model.

**Table 2 ijms-24-13724-t002:** Infants’ characteristics.

	NDM-Insulin (*n* = 8)	NDM-Diet (*n* = 13)	CONTROLS (*n* = 5)
Infants’ characteristics at birth			
Gender (male/female)	6/2	10/3	2/3
Gestational age, weeks (mean ± SD)	38 ± 3	38 ± 5	39 ± 1
Birth weight, g (mean ± SD)	3340 ± 10	3218 ± 12	3056 ± 7
Length, cm (mean ± SD)	50 ± 4	50 ± 5	49 ± 8
Head circumference, cm (mean ± SD)	34 ± 3	34 ± 1	33 ± 8

## Data Availability

The data that support the findings of this study are available from the corresponding author upon reasonable request.

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
