# Peer review of "Sardinian Infants of Diabetic Mothers: A Metabolomics Observational Study"

_ijms, 2023, doi:10.3390/ijms241813724_

Round 1

Reviewer 1 Report

This study by Dessi et al. is technically very good and the data are well-presented. Statistics are sound. The discussion explains very well the biochemical and mechanistic meaning of the alterations shown for the NDM-insulin infants. It´s interesting that the NDM-diet newborns appear to have a healthy-type of metabolome. However, it is unclear what we can conclude from that, in particular since no detailed description of the state of glucose metabolism of the mothers is given. Moreover, what is missing is an explanation, how the aim to identify new biomarkers for monitoring and early diagnosis of NDM and the predisposition to development of related chronic diseases was addressed. I don´t see this at all. The title promises some predictive assessment, but there is none. I suggest to change the title, to re-define the aims of this study and to discuss the possible diagnostic value of the findings.

Author Response

REV. 1

This study by Dessi et al. is technically very good and the data are well-presented. Statistics are sound. The discussion explains very well the biochemical and mechanistic meaning of the alterations shown for the NDM-insulin infants. It´s interesting that the NDM-diet newborns appear to have a healthy-type of metabolome. However, it is unclear what we can conclude from that, in particular since no detailed description of the state of glucose metabolism of the mothers is given. Moreover, what is missing is an explanation, how the aim to identify new biomarkers for monitoring and early diagnosis of NDM and the predisposition to development of related chronic diseases was addressed. I don´t see this at all. The title promises some predictive assessment, but there is none. I suggest to change the title, to re-define the aims of this study and to discuss the possible diagnostic value of the findings.

Thank you for your suggestions. We modified the text accordingly. Moreover, we changed the title “SARDINIAN INFANT OF DIABETIC MOTHER: A METABOLOMIC OBSERVATIONAL STUDY”. We also modified the paragraph concerning the objectives of the study, specifying that we aimed to assess the possible presence of metabolome alterations in the offspring of diabetic mothers who are insulin-treated or on diet therapy.

With regard to diagnostic potential, we have provided hypotheses regarding the metabolites with the greatest occurrence in the literature although there are only preliminary studies in this field that allow us to formulate preliminary hypothesis only. We have also tried to clarify that the discovery of altered metabolic pathways in children of diabetic mothers on insulin therapy may represent the starting point for preventive and personalized medicine that through lifestyle corrections from the earliest ages of development can reduce the incidence of chronic non-communicable diseases in predisposed individuals.

We have also included the strengths and limitations of our work.

Reviewer 2 Report

I was honored to review the manuscript. The study presents high quality and deals with important clinical issues, such type of study is needed.  I have only a few small remarks that the authors should address properly.

I recommend accepting the manuscript after minor revision.

There are only some points to correct:

 - In the “objectives” paragraph, the aim is not clearly specified, although it is understandable when reading the whole article. Could You add one clear sentence about the intention, a problem that the article is trying to solve? Maybe a hypothesis, which will be confirmed or not in the conclusion section?

 - please provide the list of abbreviations

 - please provide the number of ethical approval

  • - introduction and discussion section need improvement; please provide information on how your results will translate into clinical practice;

- in the discussion section please provide the study's strong points  and study limitation section

- please correct typos

All the abovementioned issues are crucial for the credibility of the results. The paper can be accepted only after addressing all the issues and another subsequent review.

I recommend accepting the manuscript after minor revision.

Author Response

REV. 2

I was honored to review the manuscript. The study presents high quality and deals with important clinical issues, such type of study is needed.  I have only a few small remarks that the authors should address properly.

I recommend accepting the manuscript after minor revision.

There are only some points to correct:

 - In the “objectives” paragraph, the aim is not clearly specified, although it is understandable when reading the whole article. Could You add one clear sentence about the intention, a problem that the article is trying to solve? Maybe a hypothesis, which will be confirmed or not in the conclusion section?

 - please provide the list of abbreviations

 - please provide the number of ethical approvals

- introduction and discussion section need improvement; please provide information on how your

results will translate into clinical practice;

- in the discussion section please provide the study strong points and study limitation section

- please correct typos

All the above-mentioned issues are crucial for the credibility of the results. The paper can be accepted only after addressing all the issues and another subsequent review.

I recommend accepting the manuscript after minor revision.

Thanks for your comment. We modified the paper according to your suggestions.

We have tried to define more clearly the objectives of our study, specifying the intention to investigate potential altered metabolic pathways in the offspring of diabetic mothers in order to be able to identify, also thanks to a future validation of certain biomarkers, those subjects most at risk of developing chronic non-communicable diseases. These clarifications allow us to be in line with the conclusions of our work that emphasize the importance of individualized precision medicine and early lifestyle intervention especially in the most vulnerable subjects.

Therefore, the introduction paragraph has been implemented by more clearly defining the objectives of our work and by adding some information of how our results could be translated into clinical practice in the future.

As for the discussion, we included a clearer clinical reference through metabolites with more solid or convincing literature data, although this field is still developing, and the available data are all still preliminary. We have even added the strengths and limitations of our work.

Finally, we have also included a specific list of abbreviations at the bottom of the paper and the ethics committee approval number was already in the materials and methods section, subject and sample collection.

We did our best to correct all the typos.